# Exploring the determinants of exclusive breastfeeding among infants under-six months in Ethiopia using multilevel analysis

**Shambel Aychew Tsegaw**[1], **Yeshimebet Ali Dawed**[2], **Erkihun Tadesse Amsalu** [3]*

**1** Department of Public Health, Dessie Health Science College, Dessie, Ethiopia, **2** Department of Nutrition, School of Public Health, College of Medicine and Health Sciences, Wollo University, Dessie, Ethiopia, **3** Department of Epidemiology and Biostatistics, School of Public Health, College of Medicine and Health Sciences, Wollo University, Dessie, Ethiopia

* brhaneyared07@gmail.com, erkihunt@yahoo.com

## Abstract

### Introduction

Exclusive breastfeeding (EBF) is the safest and healthiest option of feeding among infants in the first 6 months throughout the world. Thus, the promotion of EBF is essential to prevent complex infant health problems even at the adulthood level. But the majority of previous studies focused on individual- level determinants of EBF by using basic regression models in localized areas. This study aimed to identify the determinants of EBF at the individual and community level which would be helpful to design appropriate strategies for improving the practice of EBF.

### Methods

It is a secondary data analysis using the 2016 Ethiopian Demographic and Health Survey (EDHS) data. A total of 1185 infants under 6 months of age were included in the analysis. A Multilevel logistic regression model was employed to investigate factors significantly associated with EBF among under-six infants in Ethiopia. Adjusted odds ratio (AOR) with 95% confidence interval (CI) was used to measure the association of variables whereas Intracluster correlation (ICC), median odds ratio (MOR), and proportional change in variance (PCV) were used to measure random effects (variation).

### Result

In multilevel logistic regression; 4–5 months age infant (AOR = 0.04, 95%CI:0.02–0.07), female infants (AOR = 2.51, 95%CI:1.61–3.91), infant comorbidities (AOR = 0.35, 95%CI: 0.21–0.57), richest household wealth index (AOR = 10.34, 95%CI: 3.14–34.03) and antenatal care (AOR = 2.25, 95%CI:1.32–3.82) were individual- level determinants significantly associated with exclusive breastfeeding. Whereas, contextual region (AOR = 0.30, 95%CI: 0.10–0.87), community- level of postnatal visit (AOR = 2.77, 95%CI: 1.26–6.58) and community -level of maternal employment (AOR = 2.8, 95%CI: 1.21–6.47) were community level determinants significantly associated with EBF. The full model showed that46.8% of

**Data Availability Statement:** All relevant data are within the manuscript and its Supporting information files.

**Funding:** The author(s) received no specific funding for this work.

**Competing interests:** The authors have declared that no competing interests exist.

**Abbreviations:** AIC, Akaike Information Criteria; ANC, Antenatal Care; BF, Breast Feeding; BIC, Bayesian Information Criteria; CI, Confidence Interval; CSA, Central Statics Agency; DIC, Deviance Information Criteria; EBF, Exclusive Breast Feeding; EDHS, Ethiopian Demographic and Health Survey; EFMoH, Ethiopian Federal Ministry of Health; ICC, Intracluster Correlation; ICYF, Infant and Young Child Feeding; MCH, Maternal and Child Health; MOR, Median Odds Ratio; PCV, Proportional Change in Variance; PNC, Postnatal Care; VIF, Variance Inflation Factor.

the variation of exclusive breastfeeding was explained by the combined factors at the individual and community levels. Similarly, it showed that the variation in exclusive breastfeeding across communities remained statistically significant (ICC = 8.77% and variance = 0.32 with P<0.001).

## Conclusion and recommendation

Our study showed that both individual and community level determinants were significantly associated with EBF practice among under 6 infants. Based on our findings, it is recommended to promote and enhance antenatal and postnatal care services utilization of mothers to improve exclusive breastfeeding practice and more emphasis should be given to infants with comorbid conditions and those who were living in the pastoralist regions.

## Introduction

All necessary nutrients essential for survival, growth, and development, as well as immunologic, antimicrobial, and anti-inflammatory factors, are obtained from breast milk for the first six months of life [1,2]. The World Health Organization (WHO) defines exclusive breastfeeding (EBF) as when 'an infant receives only breast milk, no other liquids or solids are given not even water, except for oral rehydration solution, or drops/syrups of vitamins, minerals or medicines [3,4]. World Health Organization recommends EBF for the first six months of life, followed by continued breastfeeding with appropriate complementary foods until two years and above [5]. Breastfeeding reduces the risk of acquiring gastrointestinal (GIT) and respiratory tract infections. In later life, it also helps to prevent the risk of developing breast and uterine cancer, obesity, osteoporosis, and type 2 diabetes mellitus [4,6,7].

Globally, in 2017 around 41% of under-six infants were exclusively breastfed [8]. In low and middle- income countries, about 37% of infants were exclusively breastfed [7]. Studies also reported the magnitude of exclusive breast feeding ranged from 19% to 65% in the African context [9–12].

In the globe, more than 1.4 deaths were reported due to non-exclusive breastfeeding in the first six months of life where 41% in SSA (sub-Saharan Africa) and 34% in South Asia (SA) [3,5,8].

Improved breastfeeding practices can save about 1.5 million infants each year [8]. EBF can prevent up to 61% and 63% of admission secondary to pneumonia and diarrheal diseases respectively. It can also decrease type 1 diabetes [6].

Breastfeeding education and information during antenatal and postnatal visits were methods to improve EBF [13]. Besides, implementation of international code of breast milk marketing substitutes, community- based promotion and support, advocacy, and training were also strategies to enhance EBF practices [1].

Even if the government of Ethiopia tried to promote and implement EBF practice for infants under- six months of age but still there is a poor improvement [14]. This could be due to several contributing factors including the low level of educational status, maternal employment, operative delivery, late initiation of breastfeeding, low antenatal and postnatal visits, increased production of formula foods, and poor counseling of mother regarding EBF [10,15–18]. Furthermore, EBF practice was influenced by community- level factors including place of residence, contextual region, community level of maternal education, community level of

ANC and PNC utilization, community- level of employment, and community level of media exposure [19–21].

Despite studies on exclusive breastfeeding practice were done in Ethiopia, most of them were focused on individual level factors using a basic logistic regression model covering limited areas with a small sample size [15–21]. But EBF practice was influenced by both individual and community- level factors since the behavior of individual towards EBF practice are not only attributed by individual factors but also by community- level factors and individual infants were nested within the communities. The independence assumption among individuals within the same cluster and the equal variance assumption across clusters are violated in the case of nested data. As EDHS data has a hierarchical nature and violate the independent assumption of basic logistic regression, considering clustering effect is preferred. Thus, the current study aims to identify the determinants of EBF practice both at the individual and community level among infants undersix months of age using multilevel logistic regression analysis.

## Methods

### Study setting

This study was done in Ethiopia, which is located in the North- Eastern part of Africa. Contextually, the country is categorized as agrarian, pastoralists, and city based population. It has a total of 104,957,000 populations, of which 36,296,657 were women [22]. The Majority of the population, 83.6% living in rural areas, and 16.7% of the population reside in urban areas. The average household size at the national level is 4.7 persons [14,23]. The country has a fertility rate of 4.6, infant mortality rate (per 1,000 live births) of 48, and child mortality rate (per 1,000 live births) of 67 [14].

**Data source and population.** This study used data from the 2016 EDHS (Ethiopia Demographic and health survey) which was conducted from January 18 to June 27, 2016. Thus, a community- based cross-sectional study was employed to identify individual- level and community- level factors affecting exclusive breastfeeding among under-six month infants [14].

The source population for this study was all infants undersix months of age living in Ethiopia and all infants undersix months of age in the selected enumeration areas during the data collection period were the study population. Accordingly, a total of 1185 infants under six months of age who fulfill the eligibility criteria were included in the study using a stratified two stage cluster sampling technique. Enumeration areas (EAs) and households were the primary and secondary sampling units respectively. The detailed sampling procedure was published elsewhere [14,24].

We extracted the outcome and explanatory variables from the EDHS 2016 kids data after getting permission from Measure DHS International Program. A structure and pretested questionnaire was used for data collection during the survey. Comprehensive information about the 2016 EDHS data collection procedure has been published elsewhere [14,24,25].

**Variables of the study and operational definition.** The outcome variable of the study was Exclusive Breastfeeding (EBF) among under-six month infants in Ethiopia. Infant related variables (Age of the infant, sex of the infant, birth order, birth weight, birth interval, infant comorbidities, and time of breastfeeding initiation), maternal socio-demographic variables (age of the mother, marital status, educational status of the mother, occupational status of the mother, household wealth index, media exposure, number of underfive children, and household family size), and Obstetric and Healthcare associated variables (ANC utilization, PNC utilization, place of delivery, mode of delivery, delivery assistance, and parity) were individual-level predictors of the study. Also, community -level variables included in the study were place

of residence, contextual region, community media exposure, community wealth index, community women education, community ANC utilization, community level of employment, and community PNC utilization.

The World Health Organization (WHO) defines exclusive breastfeeding (EBF) as when 'an infant receives only breast milk, no other liquids or solids are given not even water, except for oral rehydration solution, or drops/syrups of vitamins, minerals or medicines [4]. It was measured using a 24-hour recall among mothers with infants under 6 months of age [4].

In this study, infant comorbidities were generated by aggregating variables (diarrhea, cough, fever, and shortness of breath of infants under six months of age) in the last two weeks preceding the survey, subsequently if at least one of the comorbid was found the response coded as "Yes" otherwise the response coded as "No. The occupational status of the mother was generated by using the occupation of the respondent and categorized as currently working (includes all types of work) and currently not working.

Community -level variables were generated by aggregating the individual characteristics in a cluster since EDHS did not collect data that can directly describe the characteristics of the clusters except the place of residence. The aggregates were computed using the proportion of a given variables' subcategory. Since the aggregate value for all generated variables not normally distributed, it was categorized into groups based on the national median values and based on previous related studies [24,25].

Community ANC utilization was the proportion of mothers within specific cluster who visited ANC some number of times. It was categorized using national level quartiles in to low ANC utilized community (when≤25% of women are utilizing ANC), middle (when 25–75% of women are utilizing ANC), and high (when >75% of women are utilizing ANC) [25].

The Community level of PNC utilization was the proportion of women within specific cluster who visit PNC some number of times. It was categorized as low (when≤50% of women utilized PNC) and high (when>50% of women utilized PNC) [25].

Community- level of media exposure was an aggregate respondent level of exposure for different types of media categorized as "<25% = Low", "25%-50% = Moderate" and ">50% = high media utilized communities" [25].

Community level of poverty was an aggregate wealth index categorized as "<25% = High", 25%-50% = Moderate" and ">50% = Low poverty communities" [25].

Contextual region Ethiopia is demarcated for administrative purpose into 11 regions, which are classified as an agrarian, pastoralist, and city based according to the living status of the population. The regions Tigray, Amhara, Oromia, SNNP, Gambella, and Benshangul Gumuz were categorized as agrarian. Somali and Afar regions were grouped to form pastoralist region and Harari region, Addis Ababa and Dire Dawa city administrations were grouped to form city- based populations [14,25].

Community- level of women education was the proportion of women in the community who have primary or higher education, which was categorized as low (when≤25% of women were educated), middle (when 25–75% of women were educated), and high (when >75% of women were educated) [25].

Community level employment status was the proportion of women who were employed (had to work) in the specific cluster. It was categorized as low (when≤50% of mothers were employed) and high (when>50% of mothers were employed) [24,25].

**Data management and analysis.** Sample weight was done to compensate for the unequal probability of selection between the strata that were geographically defined, as well as for non-responses. Weighing of individual interview produces the proper representation of exclusive breastfeeding and related factors. Coding, recoding, and exploratory analysis was performed. Categorization was done for continuous variables using information from different works of

literatures and re-categorization was done for categorical variables accordingly. For data analysis, STATA version 14.1 was used and descriptive statistics were used to present frequencies, with percentages in tables and using texts.

Four models were considered in the multilevel analysis to determine the model that best fits the data; Model one (Null model) without explanatory variable was developed to evaluate the null hypothesis that there is no cluster level difference in exclusive breastfeeding practice that specified only the random intercept and it presented the total variance in exclusive breastfeeding practice among clusters. Model two adjusted for an individual variable which assumes a cluster level difference of EBF practice is zero. Model three to evaluate community level factors by aggregate cluster difference of exclusive breastfeeding practice. Model four included both adjusted individual and community level factors.

The log of the probability of Exclusive breastfeeding was modeled using a two-level multilevel model as follows:

$$Log\left[\frac{\Pi ij}{1 - \Pi ij}\right] = \beta_0 + \beta_1 X_{ij} + B_2 Z_{ij} + \mu_j + e_{ij}$$

[26,27].

Where, i and j are the level 1 (individual) and level 2 (community) units, respectively; X and Z refer to individual and community-level variables, respectively; πij is the probability of exclusive breastfeeding for the i[th] infant in the j[th] community; the β's is the fixed coefficients. Whereas, β0 is the intercept-the effect on the probability of exclusive breastfeeding use in the absence of influence of predictors and *uj* showed the random effect (effect of the community on exclusive breastfeeding) for the j[th] community, and *eij* showed random errors at the individual levels. By assuming each community had a different intercept (β0) and fixed coefficient (β), the clustered data nature and the within and between community, variations were taken into account.

Multilevel logistic regression analysis was used to analyze the data since it is appropriate for DHS data as it had a hierarchical nature. Multilevel modeling was providing unexplained variation in exclusive breastfeeding due to unobserved cluster factors called the random effect. All models included a random intercept at the cluster level to capture the heterogeneity among clusters.

The measures of association (fixed-effects) estimate the association between likelihood of infants to exclusively breastfeeding as the AOR with 95% CI of various explanatory variables were expressed. Crude association between independent variables and the dependent variable was done independently and variables having p ≤0.2 in Bi-variable analysis were used to fit multivariable analysis model. At multivariable analysis,variables with p≤0.05 with confidence interval not including the null value (OR = 1)were considered as statistically significant variables with exclusive breastfeeding practice.

The measures of variation (random-effects) were reported using Intra-cluster correlation (ICC), Median Odds Ratio (MOR), and Proportional Change in Variance (PCV). ICC was used to explain cluster variation while MOR is a measure of unexplained cluster heterogeneity [27]. The ICC shows the variation in exclusive breastfeeding of infants undersix months of age due to community characteristics. The higher the ICC (ICC>5%), the more relevant was the community characteristics for understanding individual variation in exclusive breastfeeding of infants. The ICC can be calculated as follows: $\left[ICC = \frac{\delta^2 u}{\delta^2 u + \delta^2 e}\right]$ where $\delta^2 u$ = between group variation, $\delta^2 e$ = with in group variation OR $\left[ICC = \frac{\delta^2}{\delta^2 + \frac{\pi^2}{3}}\right]$, where $\delta^2$ is the estimated variance of clusters [26]. The STATA software command can also compute the ICC value of each model.

MOR is defined as the median value of the odds ratio between the area at highest likelihood and the area at the lowest likelihood of exclusive breastfeeding when randomly picking out two areas and it measures the unexplained cluster heterogeneity; the variation between clusters by comparing two persons from two randomly chosen different clusters. MOR can be calculated using the formula

$$[\text{MOR} = \exp(\sqrt{2x\delta^2 + 0.6745}) \approx \exp(0.95\delta)]$$

[26].

In this study, MOR shows the extent to which the individual probability of being exclusively breastfed is determined by residential area. The proportional change in variance [PCV = (*VA* − *VB*)/*VA*) * 100] where VA = Variance of initial model and VB = Variance of model with more terms measures the total variation attributed by individual level and community level factors in the multilevel model [26]. PCV was computed for each model concerning the empty model as a reference to show power of the factors in the model explains exclusive breastfeeding practice.

Log- likelihood test, Deviance Information Criteria (DIC), and Akaike Information Criteria (AIC) were used to estimate the goodness of fit of the adjusted final model in comparison to the preceding models (individual and community level models). Thus, the model with the highest value of Log likelihood test and with lowest values of DIC and AIC was considered to be the best fit model.

**Ethical consideration.** Ethical clearance was obtained from the Ethical Review Committee of the College of Medicine and Health Sciences, Wollo University with approval and supporting letter. Permission to access the data set was obtained from the Measure DHS International Program. The data was only used for purpose of this study and not shared with a third party. All data used in this study were anonymous publicly available and aggregated secondary data with not having any personal identity. The data was fully available on the full DHS website (www.measuredhs.com).

## Results

### Infant characteristics

A total of 1092 infants under six months of age were included in the 2016 EDHS survey. But after weighting the sample size became 1185, and all of them were included in the analysis. The mean age of respondents was 2.54 months (SD ± 0.51). The highest proportion of infants, about 418(35.26%) were in the age group of 4–5 months and 605(51.07%) were females. Regarding breastfeeding initiation majority, about 817 (68.95%) of the infants have initiated breastfeeding immediately after birth (Table 1).

### Maternal socio-demographic and socioeconomic factors

Among 1185 infants under six months in the 2016 EDHS survey, the highest proportion, about 573 (48.41%) of the mothers were in the age group of 25–34 years old. Regarding the educational status of the mothers, more than half, 698(60.9%) of them had no formal education. Similarly, the majority of mothers, about 1141(96.3%) of them were in a marital union at the time of the survey.

More than half of mothers about, 738(62.28%) of them were not working and about 781 (65.91%) of them had no media exposure about exclusive breastfeeding. The highest proportion of mothers, about 291(24.56%) of them were from the poorest household wealth index (Table 2).

**Table 1. Characteristics of infants under six months of age in Ethiopia, 2016 (N = 1185).**

| Variables | Categories | Weighted frequency (%) | EBF prevalence | |
|---|---|---|---|---|
| | | | No (%) | Yes (%) |
| Age of infant | 0–1 month | 388 (32.78) | 101(25.78) | 288(74.22) |
| | 2–3 months | 378(31.96) | 136(35.89) | 243(64.11) |
| | 4–5 months | 418(35.26) | 267(64.02) | 150(35.98) |
| Sex of infant | Male | 580(48.93) | 248(42.75) | 331(57.25) |
| | Female | 605(51.07) | 255(42.14) | 350(57.85) |
| Birth interval | No previous birth | 278(23.43) | 137(49.28) | 141(50.71) |
| | <24 months | 118(9.92) | 53(45.29) | 64(54.71) |
| | ≥24 months | 790(66.65) | 314(39.74) | 476(60.26) |
| Birth order | First | 278(23.43) | 137(49.45) | 141(50.54) |
| | 2–3 | 335(28.27) | 131(39.10) | 204(60.89) |
| | 4$^+$ | 572(48.29) | 236(41.25) | 336(58.74) |
| Infant Comorbidities* | Yes | 296(24.99) | 162(54.72) | 134(45.27) |
| | No | 885(74.64) | 342(38.64) | 543(61.36) |
| TimeofBF initiation** | Immediatelyafter birth | 817(68.95) | 317(38.80) | 500(61.20) |
| | Within hours | 284(23.92) | 126(44.36) | 158(55.64) |
| | Within days | 60(5.06) | 36(60.00) | 24(40.00) |
| Birth weight | Low | 24(2.02) | 14(58.34) | 10(41.66) |
| | Normal | 165(13.93) | 69(41.82) | 96(58.18)) |
| | Macrosomia | 21.27(1.80) | 10(47.62) | 11(52.38) |
| | Didn't weighted | 841(70.97) | 365(43.40) | 476(56.6) |
| | Don't know | 134(11.24) | 46(34.32) | 88(65.68) |

* Infant comorbidities missing (N = 4),

**Breastfeeding initiation missing (N = 24).

## Obstetric and healthcare -related factors

Regarding antenatal care visit the majority, about 770(64.95%) of mothers had ANC visits. Similarly, more than half of mothers, about 732(61.8%) of them were delivered at home. About 35 (2.95%) of mothers were delivered by caesarian section and more than half of mothers 74(6.23%) had PNC visit in the last 2 months before the survey (Table 3).

## Community -level factors

In our study, most of the respondents, about 1048(88.44%) of them were rural residents and about 1086(91.66%) of them were from agrarian regions. The highest proportion, about 495 (41.77%) of respondents were from low poverty level of the community and about 840 (70.89%) of them had low community media utilization. Regarding community postnatal care utilization, 1168 (98.56%) of the mothers had low community postnatal care utilization. Similarly, the majority of the mothers, about 840 (70.85%) and 838(70.73%) belong to low community educational level and low community employment status respectively (Table 4).

## Determinants of exclusive breastfeeding practice

**Fixed effects (Measures of association).** Bi-variable multilevel logistic analysis computed and p-value up to 0.2 was selected to fit the multi-variablemultilevel logistic regression model to control confounding. Accordingly, Age of infant, sex of infant, birth weight, birth interval, birth order, infant comorbidities, breastfeeding initiation, marital status, maternal occupation,

**Table 2. Maternal socio-demographic and socioeconomic characteristics in Ethiopia, 2016 (n = 1185).**

| Variables | Categories | Weighted frequency (%) | EBF prevalence | |
|---|---|---|---|---|
| | | | No (%) | Yes (%) |
| Age of the mother (years) | 15–24 | 397(33.49) | 166(41.81) | 231(58.19) |
| | 25–34 | 573(48.41) | 246(42.85) | 328(57.15) |
| | 35–49 | 215(18.11) | 92(42.99) | 122(57.01) |
| Marital status | Not in union | 44(3.70) | 22(50.00) | 22(50.00) |
| | Currently in union | 1141(96.30) | 482(42.24) | 659(57.76) |
| Maternal education | No formal education | 698(60.96) | 298(42.69) | 400(57.31) |
| | Primary | 368(31.01) | 152(41.30) | 216(58.70) |
| | Secondary and above | 119(10.07) | 54(45.37) | 65(54.63) |
| Maternal occupation | Currently Not working | 738(62.28) | 332(44.98) | 406(55.01) |
| | Currently Working | 447(37.72) | 172(38.47) | 275(61.53) |
| household family size | <5 | 312(26.31) | 130(41.66) | 182(58.34) |
| | 5–6 | 401(33.87) | 153(38.15) | 248(61.85) |
| | 7–8 | 320(27.03) | 150(46.88) | 170(52.12) |
| | ≥9 | 152(12.78) | 70(46.35) | 81(53.65) |
| Number of under five children | 0–1 | 401(33.87) | 168(41.89) | 233(58.10) |
| | 2 | 504(42.52) | 203(40.27) | 301(49.73) |
| | 3 | 278(23.61) | 133(47.50) | 147(52.50) |
| Wealth index | Poorest | 291(24.53) | 131(45.01) | 160(54.09) |
| | Poorer | 277(23.40) | 101(36.46) | 176(63.54) |
| | Middle | 217(18.35) | 98(45.16) | 119(54.84) |
| | Richer | 215(18.18) | 114(52.77) | 102(47.23) |
| | Richest | 184(15.53) | 61(33.15) | 123(66.84) |
| Media exposure | No | 781(65.92) | 324(41.48) | 457(58.51) |
| | Yes | 404(34.08) | 180(44.55) | 224(55.45) |

**Table 3. Maternal obstetric and healthcare-related characteristics in Ethiopia, 2016 (n = 1185).**

| Variables | Categories | Weighted N (%) | EBF prevalence | |
|---|---|---|---|---|
| | | | No (%) | Yes (%) |
| Parity | Nulliparous | 278(23.43) | 137(49.3) | 141(50.7) |
| | Multiparous | 907(76.57) | 367(40.46) | 540(59.53) |
| ANC visit* | No | 412 (34.80) | 195(47.33) | 217(52.67) |
| | Yes | 770 (64.95) | 309(40.12) | 461(59.87) |
| Place of delivery | Home | 732 (61.80) | 318(43.44) | 414(56.56) |
| | Health institution | 453 (38.20) | 186(41.05) | 267(58.95) |
| Caesarian delivery | No | 1150 (97.06) | 486(42.26) | 664(57.74) |
| | Yes | 35(2.94) | 18(51.42) | 17(48.58) |
| Delivery assistance | Professionals | 741(62.52) | 323(43.58) | 418(56.42) |
| | TBAs | 444(37.48) | 181(40.76) | 263(59.23) |
| PNC visit | No | 1111(93.77) | 465(41.85) | 646(58.15) |
| | Yes | 74(6.23) | 39(52.70) | 35(47.29) |

*ANC follow up missed (N = 3).

**Table 4. Community- level determinants of exclusive breastfeeding in Ethiopia, 2016 (N = 1185).**

| Variables | Categories | Weighted frequency N (%) | EBF prevalence | |
|---|---|---|---|---|
| | | | No (%) | Yes (%) |
| Residence | Urban | 137(11.56) | 58(42.33) | 79(57.67) |
| | Rural | 1048(88.44) | 446(42.55) | 602((57.45) |
| Contextual region | Agrarian | 1086(91.66) | 451(41.52) | 635(58.47) |
| | Pastoralist | 65(5.50) | 38(58.46) | 27(41.54) |
| | City based | 34(2.84) | 15(44.11) | 19(55.89) |
| Community ANC utilization | Low | 253(21.37) | 109(43.09) | 144(56.91) |
| | Moderate | 719(60.70) | 304(42.29) | 415(57.71) |
| | High | 213(17.93) | 112(43.07) | 148((56.92) |
| Community PNC utilization | Low | 1168(98.56) | 498(42.63) | 670(57.37) |
| | High | 17(1.44) | 6(35.30) | 11(64.70) |
| Community poverty level | High | 344(29.01) | 143(41.70) | 200(58.30) |
| | Moderate | 346(29.22) | 149(43.06) | 197(56.93) |
| | Low | 495(41.77) | 211(42.63) | 284(57.37) |
| Community media utilization | Low | 840(70.89) | 346(41.19) | 494(58.80) |
| | Moderate | 217(18.29) | 96(44.23) | 121(55.76) |
| | High | 128(10.82) | 63(49.22) | 65(50.78) |
| Community women education | Low | 840(70.85) | 359(42.73) | 481(57.26) |
| | Moderate | 244(20.62) | 90(36.89) | 154(63.11) |
| | High | 101(8.52) | 55(54.45) | 46(45.54) |
| Community maternal employment status | Low | 838(70.73) | 377(44.99) | 461(55.01) |
| | High | 347(29.27) | 127(36.70) | 220(63.40) |

In this study, the prevalence of exclusive breastfeeding among infants undersix months of age was 58% with a 95% CI of (54.6, 60.2).

maternal education, household family size, number of under-five children, wealth index, parity, ANC visit, caesarian delivery, PNC visit, residence, contextual region, community ANC utilization, community PNC utilization, community media exposure, community level of poverty, community level of employment status and community level of women education pass Bi-variable multilevel logistic regression model at p- value≤ 0.2.

Four models were fitted hierarchically in this multi-level logistic regression analysis. Model one or Null model (without predictors), model two (only individual- level factors), model three (only community- level factors) and model four (both individual and community level factors). Finally, p-value ≤ 0.05 and odds ratio not including the null value (OR = 1) was used to select variables that had a statistically significant association with exclusive breastfeeding practice in the final model.

As shown in Table 5 below, our study finding showed that age of infants, sex of infants, infant comorbidities, wealth index, antenatal care visit, contextual region, community postnatal care visit, and community level of employment status were statistically significant determinants of exclusive breastfeeding in the final model.

The age of the infant was negatively associated with exclusive breastfeeding. Those infants whose age group was between 2–3 months were 60.9% [AOR = 0.39, 95% CI: (0.23, 0.65)] and 4–5 months were 95.8% [AOR = 0.04, 95% CI: (0.02, 0.07)] less likely to exclusive breastfeed as compared to infants whose age was up to one month.

The sex of the infant was significantly associated with exclusive breastfeeding. Female infants were 2.5 [AOR = 2.51, 95% CI: (1.61, 3.91)] times more likely to exclusively breastfeed as compared to male infants.

**Table 5. Multilevel logistic regression analysis of both individual and community level factors associated with exclusive breastfeeding among infants under 6 months of age in Ethiopia, 2016 (N = 1185).**

| Variables | Categories | Model one | Model two AOR (95%CI) | Model three AOR (95%CI) | Model four AOR (95%CI) |
|---|---|---|---|---|---|
| Age of infant | 0–1 month | | 1 | | 1 |
| | 2–3 months | | **0.34(0.21,0.56)**\*\*\* | | **0.39(0.23, 0.65)** \*\*\* |
| | 4–5 months | | **0.05(0.02,0.08)**\*\*\* | | **0.042(.02, 0.07)** \*\*\* |
| Sex of infant | Male | | 1 | | 1 |
| | Female | | **2.43(1.61,3.65)** \*\*\* | | **2.51(1.61, 3.91)** \*\*\* |
| Birth interval | No previous birth | | 1 | | 1 |
| | <24 months | | **4.27 (1.36, 13.39)**\* | | 3.44(0 1.19, 9.92) |
| | > = 24 months | | 2.73(1.00, 7.41) | | 1.84(0.74, 4.54) |
| Birth order | First | | 1 | | 1 |
| | 2–3 | | 2.71(0.99, 7.34) | | 1.821(0.15, 2.86) |
| | 4⁺ | | 3.32(0.93,11.87) | | 1.46(0.97, 2.18) |
| Infant comorbidities | No | | 1 | | 1 |
| | Yes | | **0.34(0.19, 0.56)**\*\*\* | | **0.35(0 .21, 0.57)**\*\*\* |
| Time of BF initiation | Immediately after birth | | 1 | | 1 |
| | Within hours | | 1.05(0.63, 1.76) | | 1.96(0 .68, 5.65) |
| | Within days | | 0.57(0.19, 1.71) | | 2.02(0 .66, 6.15) |
| Birth weight | Low | | 1 | | 1 |
| | Normal | | 4.40(0.67,28.74) | | 2.91(0.49, 17.20) |
| | Macrosomia | | 2.72(0.23, 31.39) | | 2.28(0.21, 24.30) |
| | Didn't weighted | | 2.64(0.39, 17.77) | | 1.86(0.30, 11.49) |
| | Don't know | | 2.05(0.30, 13.95) | | 1.67(0 .271, 10.360) |
| Marital status | Not in union | | 1 | | 1 |
| | Currently in union | | 2.36(0.93, 6.04) | | 1.98(0.75, 5.21) |
| Maternal education | No formal education | | 1 | | 1 |
| | Primary | | 1.18(0.64, 2.17) | | 1.50(0.84, 2.70) |
| | > = Secondary | | 0.50(0.15,1.66) | | 0.90(0.30, 2.69) |
| Maternal occupation | Currently not working | | 1 | | 1 |
| | Currently Working | | 0.58(0.32,1.02) | | 0.65(0 .37, 1.12) |
| Number of household members | <5 | | 1 | | 1 |
| | 5–6 | | 2.20(1.11,4.36) | | 0.84(0 .36, 1.97) |
| | 7–8 | | 0.61(0.26,1.42) | | 1.27(0.623, 2.60) |
| | > = 9 | | 1.16(0.427,3.16) | | 0.48(0.23, 0.96) |
| Number of under five children | 0–1 | | 1 | | 1 |
| | 2 | | 1.052(0.49,2.22) | | 1.65(0.72, 3.82) |
| | > = 3 | | 0.876(0.36,2.09) | | 1.52(0 .86, 2.69) |
| Wealth index | Poorest | | 1 | | 1 |
| | Poorer | | **1.99(1.01, 3.93)**\* | | 1.72(0.90, 3.31) |
| | Middle | | 1.33(0.58, 3.05) | | 1.37(0 .61, 3.07) |
| | Richer | | 0 .90(0.34,2.36) | | 0.94(0.37, 2.39) |
| | Richest | | **15.79(4.2, 59.39)**\*\*\* | | **10.34(3.14, 34.03)**\*\*\* |
| Parity | Nuliparous | | 1 | | 1 |
| | Multiparous | | **1.61(1.11, 2.33)**\* | | 2.21(0.95, 5.11) |
| ANC visit | No | | 1 | | 1 |
| | Yes | | **2.75(1.53, 4.94)**\*\* | | **2.25(1.32, 3.82)**\*\* |
| Caesarian delivery | No | | 1 | | 1 |
| | Yes | | 0.53(0.14, 2.04) | | 0.47(0 .128, 1.73) |

*(Continued)*

**Table 5.** (Continued)

| Variables | Categories | Model one | Model two AOR (95%CI) | Model three AOR (95%CI) | Model four AOR (95%CI) |
|---|---|---|---|---|---|
| PNC visit in 2 months | No | | 1 | | 1 |
| | Yes | | **0.10(0.03, 0.30)** * | | 0.15(0 .054, 1 .45) |
| Residence | Urban | | | 1 | 1 |
| | Rural | | | 0.85(0.36, 2.027) | 3.414(0 .82, 14.109) |
| Contextual region | Agrarian | | | 1 | 1 |
| | Pastoralist | | | **0.38(0.17, 0.85)*** | **0.30(0 .10, 0.86)*** |
| | City based | | | 0.99(0.32 3.00) | 0.56(0 .12, 2.52) |
| Community ANC utilization | Low | | | 1 | 1 |
| | Moderate | | | 0.97(0.51, 1.87) | 1.07(0 .56, 2.02) |
| | High | | | 1.09(0.46, 2.56) | 1.30(0 .62, 2.71) |
| Community PNC utilization | Low | | | 1 | 1 |
| | High | | | 1.42(0.34, 5.97) | **2.77(1.26, 6.58)*** |
| Community poverty level | High | | | 1 | 1 |
| | Moderate | | | 0.79(0.39, 1.60) | 0.97(0.33, 2.84) |
| | Low | | | 0.83 (0.42,1.62) | 0.99(0.31, 3.10) |
| Community media utilization | Low | | | 1 | 1 |
| | Moderate | | | 0.83(0.45, 1.53) | 0.86(0 .351, 2.13) |
| | High | | | 0.60(0.26, 1.42) | 0.84(0.23, 3.02) |
| Community women education | Low | | | 1 | |
| | Moderate | | | 1.16(0.63, 2.14) | 1.13(0 .454, 2.83) |
| | High | | | 0.52 (0.22,1.21) | 0.53(0.13, 2.06) |
| Community employment status | Low | | | 1 | 1 |
| | High | | | 1.536(0.91, 2.59) | **2.81(1.207, 6.47)*** |

Model one: empty model, CI: Confidence Interval, AOR: Adjusted Odds Ratio, 1: Reference category *p<0.05 **p<0.01 ***p<0.001.

Infant comorbidities were negatively associated with exclusive breastfeeding. Those infants who had comorbidities were 66% [AOR = 0.34, 95%CI: (0.21, 0.57)] less likely to exclusively breastfeed as compared to infants who had no comorbidities.

Household wealth index was positively associated with exclusive breastfeeding. Those infants from the richest family wealth index were 10 times [AOR = 10.34, 95% CI: (3.14, 34.03)] more likely exclusive breastfeeding as compared to those infants from the poorest family wealth index.

The maternal antenatal visit was also positively associated with exclusive breastfeeding. Those infants whose mothers had ANC visits were 2.25 times [AOR = 2.25, 95%CI: (1.323, 3.82)] more likely to exclusively breastfeed compared to those infants whose mothers had no ANC visits.

The contextual region was significantly associated with exclusive breastfeeding practice. Consequently, infants who live in pastoralist regions were 70% less likely to exclusively breastfeeding [AOR = 0.30, 95% CI: (0.104, 0.86)] as compared to those infants who live in the agrarian regions.

Community level postnatal care utilization was significantly associated with EBF. As a result, those infants who live in community who had high level of PNC utilization were 2.8 [AOR = 2.77, 95% CI: 1.26, 6.58)] times more likely to exclusively breastfeed compared to infants who live in the community who had a low level of PNC service utilization.

Community level employment status was positively associated with exclusive breastfeeding practice. Thus, infants who live in a community with a high level of employment were 2.8 [AOR = 2.81, 95% CI: (1.20, 6.47)] times more likely to exclusively breastfed compared to those infants who live in a community with a low level of employment.

**Random effects (Measures of variation).** This study also aimed to show if the characteristics of the clusters where under six infants resided would affect exclusive breastfeeding practice.A Model with the lowest DIC, the lowest AIC, and the highest log- likelihood ratio was selected which was model four that better explain exclusive breastfeeding practice. At the empty model, the ICC was 0.141 (ICC>5%) and the variance was 0.54 with 95% CI: 0.25–0.75) at a p-value of <0.001, which indicates about 14.1% of the variation in exclusive breastfeeding was linked to community-level factors and there was a significant variation in exclusive breastfeeding practice across the communities (clusters).

The full model, after adjusting for individual and community level factors, showed that the variation in exclusive breastfeeding across communities remained statistically significant (ICC = 8.77% and variance = 0.32).

In this study, the full model showed up with higher PCV (46.8%); that is, 46.8% of variation of exclusive breastfeeding was explained by the combined factors at the individual and community levels. The median odds ratio of the null model (MOR = 2.01) and the final model (MOR = 1.71) were significant. The MOR at a cluster where a high proportion of nonexclusive breastfeeding was 2.01 times higher compared to a cluster with a low proportion of non- exclusive breastfeeding at the null model, whereas the MOR at a cluster where a high proportion of non-exclusive breastfeeding was 1.71 times higher compared to a cluster with a low proportion of non-exclusive breastfeeding at the final model. This indicates that 0.3 (30%) of the heterogeneity was explained by both individual and community level factors, but still there are a residual effect not explained by individual and community level variables at the final full model (MOR = 1.71) (Table 6).

## Discussion

The age of the infant was negatively associated with exclusive breastfeeding. This finding was similar with studies done in Bahr- Dar, Hawassa, Jigjiga, Zimbabwe, and India [19,20,25,28,29]. The possible justification could be as the age of the infant increases mothers initiate additional food due to the perception that breast milk alone is not enough to meet the water and nutritional demands of their infants. Another possible explanation may be related

**Table 6. Measure of variations and model fitness.**

| Characters | | Model 1 | Model 2 | Model 3 | Model 4 |
|---|---|---|---|---|---|
| Random effect | Variance | 0.54 | 0.45 | 0.31 | 0.32 |
| Measure of variation | ICC | 14.08% | 12.13% | 8.64% | 8.77% |
| | PCV | Reference | 20.77% | 42.3% | 46.8%* |
| | MOR | 2.01 | 1.83 | 1.69 | 1.71* |
| Model diagnostics | Log-likelihood | -750.82 | -593.21 | -745.80 | -571.89* |
| | DIC | 1501.64 | 1186.42 | 1491.6 | 1183.78* |
| | AIC | 1505.64 | 1248.41 | 1511.60 | 1235.77* |

*Model four is selected based on fitted statistics of log-likelihood ratio.

**Key:** Model 1 (null model) = without independent variables, Model 2 = only individual- level variables, Model 3 = only community- level variables, Model 4 = both individual and community level variables, PCV = Proportional Change in Variance, ICC = Intra Class Correlation, MOR = Median Odds Ratio, AIC = Akaike Information Criteria, DIC = Deviance Information Criteria.

to the perception of mothers that breast milk production decreased through time which is not sufficient for the growth of the infant. In addition, it could be related to poor knowledge of mothers about the importance of EBF and the adverse consequences of initiation of complementary feeding before six months of age. Furthermore, employed mothers returned to their work as the age of the infant increases which have no enough time to exclusive breastfeeding their infants [19,20,25,28].

The sex of the infant was also one of the individual level factors significantly associated with exclusive breastfeeding. Being female infants was positively associated with exclusively breastfeeding. The finding was consistent with studies done in Kenya, Cameroon, Angola, and Ghana [10,30–32]. The study showed that female infants were more likely to be exclusively breastfed compared to male infants i.e. male infants were more likely to start complementary feeding earlier compared to females. This may be related with the perception that breast milk alone does not meet their nutritional requirements and the belief that male infants have a more voracious appetite and needs additional food intake than female infants that leads the early initiation of complementary foods for male infants. The other reason may be the belief that female infants were considered as good breast suckers so breast milk alone may be enough to satisfy their feeding requirement so that additional feeding may not be initiated early [10,30,33].

Infant comorbidity was negatively associated with exclusive breastfeeding. This is concurrent with studies done in Bahr-Dar [34] and Egypt [9] that showed infants with comorbidities were less likely to be exclusively breastfed compared to healthy infants. The possible justification might be when the infant became sick; mothers may perceive that an additional diet is essential to boost the energy and immunity of the infant. Furthermore, they may perceive additional foods may also be used as a treatment option for the sick infant [9]. This implies that close follows up of the sick infant is essential to improve EBF practice.

Similarly, household wealth index showed a positive association with EBF. This is similar with studies done in Ghana, India, and Australia [17,20,35]. The possible reasons may be increased uptake of breastfeeding related information and better skills in negotiating flexible workplace hours even can stay at home that creates opportunities for exclusive breastfeeding [20]. Similarly, the low practice of EBF among the poorest wealth index could be related to poor awareness of EBF practice, stressful living situation to overcome the hardship of living. In addition, it can also be suggested that mothers from the poorest wealth index group might consider themselves as producing inadequate breast milk to satisfy their infant's demand that makes them initiate additional foods [17]. Thus, mothers in the richest wealth index families do not need to go work during lactation so they have enough time to breastfeed their infants [36].

Among individual level factors, maternal antenatal care visit was also positively associated with EBF. This finding was concurrent with studies done in India, Ghana, and different parts of Ethiopia [17,19–21,30,37]. The possible justification could be that antenatal visit is the appropriate time for breastfeeding education and information that may improve EBF later after the birth of the infant. Accordingly, breastfeeding counseling during antenatal periods significantly improves exclusive breastfeeding practice [19,25,37]. This implies that antenatal care coverage must be enhanced accordingly to improve EBF practice.

The contextual region, community level of PNC utilization, and community level of employment status were community level determinants of exclusive breastfeeding practice among infants under six months of age in Ethiopia.

In this study, contextual region was significantly associated with EBF practice. Infants who live in pastoralist regions were less likely to exclusive breastfeeding as compared to infants found in agrarian regions. The regional variation of exclusive breastfeeding was also observed

in previous studies in Ghana [17] and Malawi [38]. The variation in this study could be related that pastoralists were not well informed about the importance of exclusive breastfeeding due to the weak health care system and low women empowerment in the area. In addition, pastoralists have a mobile type of life style so they might not give attention to EBF of their infants rather they may initiate cow milk early [25]. Furthermore, there may be regional differences in some background characteristics such as culture, religion, living condition, availability and accessibility of maternal and child health services.

Community level postnatal care utilization was positively associated with EBF practice. This finding was concurrent with studies done in India, Ghana, and different parts of Ethiopia [16,17,19,37,39]. The possible explanation may be that postnatal care utilization is the best opportunity to increase the knowledge and attitude of mothers towards exclusive breastfeeding through counseling and health education since it is the appropriate period of counseling and educating mothers about essential feeding practices including EBF. Thus, as the number of women who visited postnatal care in the community increases the more likely to practice exclusive breastfeeding [18,19].

In the current study, the community level of maternal employment status was positively associated with EBF practice. This may be due to the reason that the employed community was at a higher educational level that lead them to have good information exchange about benefits of EBF through different media. In addition, an arrangement of breastfeeding time for the employees was one of the alarming mechanisms that may lead to improve EBF. The other reason may be the promotion and implementation of maternity leave according to international labor organization conventions and support of working mothers to exclusively breastfeed until six months of age [40]. But this finding is contrary to studies done in Bahr Dar [34], Hawassa [28], Tanzania [41], and Australia [20]. This may be due to the reason that studies might be done before the implementation of proclamations for maternal leave and they may also use breast milk substitutes early before six months to compensate for infant feeding during working time.

As strength the study used nationally representative data and conducted using a multilevel approach to identify individual and community level determinants of EBF that helps to provide information to design interventions strategies. Additionally, appropriate estimation adjustments like weighting and accounting for sample design were applied for analysis to represent the national population. Despite its strengths, this study has its limitations. The data used for this analysis were from a cross sectional survey, consequently, only associations were examined and it was difficult conclude about causality and it also brought recall bias. This study used a 24- hour recall method to measure exclusive breastfeeding and also the participants were infants under six months of age, no evidence to continue exclusive breastfeeding after the survey. In addition, this study was limited only to the variables collected by EDHS; some important variables were not included like health status of the mother, maternal knowledge and attitude towards exclusive breastfeeding. Furthermore, community level factors of exclusive breastfeeding were not addressed previously which makes it difficult to compare and contrast the study findings.

## Conclusion

In this study, both individual and community- level variables were significantly associated with exclusive breastfeeding among infants undersix months in Ethiopia. Thus, age of the infants, sex of the infants, infant comorbidities, wealth index and antenatal care visit were individual level factors significantly associated with exclusive breastfeeding. Accordingly, being female sex, the richest house hold wealth index, and ANC visit associated with increased

exclusive breastfeeding practice, whereas, age of the infant and infant comorbidities showed a negative association with exclusive breastfeeding. The contextual region, community level of PNC utilization, and community level of employment status were community level variables that showed statistically significant association with exclusive breastfeeding. Thus, infants from a higher level of community PNC utilization and employment status associated with increased exclusive breastfeeding, whereas, infants from pastoralist regions were less likely to exclusive breastfeeding.

## Implications of the study

Exclusive breastfeeding is one of the core indicators of infant and young child feeding, among strategies of reducing infant mortality and morbidity. It determines future growths and developments of the infants both physical and mental. The study demonstrates that addressing individual and community- level factors associated with exclusive breastfeeding practice through policies and programs was essential to improve exclusive breastfeeding practice. Therefore, emphasis should be given on encouraging women to have ANC and PNC follow- ups, where they may get information and education that will improve exclusive breastfeeding. The infants with comorbid conditions need attention since they have low exclusive breastfeeding tendency. Since maternal employment and wealth status have a higher likelihood of exclusive breastfeeding, empowering women both economically and in their employment status is implicated. Therefore, interventions on individuals and community levels were demanded saving the lives of the infants and reducing economic losses of a country.

## Supporting information

**S1 Dataset.**
(DTA)

## Acknowledgments

We would like to thank the measure DHS program for their permission to access the data. We would also acknowledge all individuals who are involved in the accomplishment of this work.

## Author Contributions

**Conceptualization:** Shambel Aychew Tsegaw.

**Data curation:** Erkihun Tadesse Amsalu.

**Formal analysis:** Shambel Aychew Tsegaw, Yeshimebet Ali Dawed, Erkihun Tadesse Amsalu.

**Investigation:** Shambel Aychew Tsegaw, Yeshimebet Ali Dawed, Erkihun Tadesse Amsalu.

**Methodology:** Shambel Aychew Tsegaw, Yeshimebet Ali Dawed, Erkihun Tadesse Amsalu.

**Software:** Yeshimebet Ali Dawed, Erkihun Tadesse Amsalu.

**Validation:** Erkihun Tadesse Amsalu.

**Writing – original draft:** Shambel Aychew Tsegaw, Yeshimebet Ali Dawed, Erkihun Tadesse Amsalu.

**Writing – review & editing:** Shambel Aychew Tsegaw, Yeshimebet Ali Dawed, Erkihun Tadesse Amsalu.

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
