## [Decision Letter · Decision Letter 0]

28 Sep 2020

PONE-D-20-23410

Exploring Determinants of Exclusive Breast feeding among infants under- six months in Ethiopia using multilevel analysis

PLOS ONE

Dear Dr. Tadesse,

Thank you for submitting your manuscript to PLOS ONE. After careful consideration, we feel that it has merit but does not fully meet PLOS ONE’s publication criteria as it currently stands. Therefore, we invite you to submit a revised version of the manuscript that addresses the points raised during the review process.

We look forward to receiving your revised manuscript.

Kind regards,

Mohammad Rifat Haider, MBBS, MHE, MPS, PhD

Academic Editor

PLOS ONE

Journal Requirements:

3. Thank you for including your ethics statement: 

"Ethical clearance was obtained from the Ethical Review Committee of College of Medicine and Health Sciences, Wollo University with approval and supporting letter. Permission to access the data set was obtained from Measure DHS International Program. The data was only used for purpose of this study and not shared to the third party.".   

i) Please provide additional details regarding participant consent. In the ethics statement in the Methods and online submission information, please ensure that you have specified (1) whether consent was informed and (2) what type you obtained (for instance, written or verbal, and if verbal, how it was documented and witnessed). If your study included minors, state whether you obtained consent from parents or guardians. If the need for consent was waived by the ethics committee, please include this information.

ii) Once you have amended this/these statement(s) in the Methods section of the manuscript, please add the same text to the “Ethics Statement” field of the submission form (via “Edit Submission”).

Additional Editor Comments (if provided):

Take care of the comments mentioned by the reviewers and especially take care of the language and grammar.

Reviewers' comments:

Reviewer's Responses to Questions

**Comments to the Author**

1. Is the manuscript technically sound, and do the data support the conclusions?

Reviewer #1: Partly

Reviewer #2: Yes

Reviewer #3: Yes

2. Has the statistical analysis been performed appropriately and rigorously? 

Reviewer #1: No

Reviewer #2: Yes

Reviewer #3: Yes

3. Have the authors made all data underlying the findings in their manuscript fully available?

Reviewer #1: Yes

Reviewer #2: Yes

Reviewer #3: Yes

4. Is the manuscript presented in an intelligible fashion and written in standard English?

Reviewer #1: Yes

Reviewer #2: Yes

Reviewer #3: No

5. Review Comments to the Author

Reviewer #1: 1. Definition of EBF should be added except allow of oral re-hydration saline

2. Line 165 the tense should not be future tense (will be)

3. Line 178 categorizing poverty is not enough; you need to mentioned how you have calculated the wealth index with what were the variables name

4. Line 224: Multi level logistic regression should be three outcome variables need to be mentioned in your write up

5. Birth weight: In survey measuring the birth weight is very uncommon. You should describe how you have collected birth weight. This is also important for others countries

6. You should consider only 4 ANC that is the current indicator for global comparison

7. Need to provide a multi-level logistic regression table otherwise description is not enough

Reviewer #2: Generally, the paper is very smart, attractive and done based on scientific approach. Investigating this issue is at this time is so important and it’s really appreciated. However, I have few comments to be considered by the authors.

Reviewer #3: This study is conducted to examine the association of exclusive breastfeeding with individual and community level factors using the 2016 EDHS survey. This survey is nationally representative and the authors used appropriate analysis which obviously enough to reach a strong conclusion. Considering these the paper must demand for publication. However I am concerned about the writing quality of this paper, ways of presenting information, confounder selection and the study discussion. I have mentioned a fews- I hope the authors would be careful about similar types of errors throughout the manuscript if they get revision from the academic editor. 

Abstract 

 In the first three lines of the abstract, authors talked about EBF and from the fourth line they started talking about determinants of EBF. They are different, however, presented tufter here without logical link. 

In line 24-25 of the abstract as well as in the study title, the authors talked about individual and community  level factors. When I started reading this manuscript, I had a feeling that this study about individual and community level determinants of EBF. I am sure other readers will have the same feeling by seeing the study title. However, in actual practice, authors considered many determinants including individual level factors, household level factors, community level factors and child level factors. I would recommend authors to revise their title so that nobody would have the feeling by seeing the title that I had in my very first look. 

Line 28-29 was unclear 

Line 32, you abbreviated exclusive breastfeeding before in the first line, why do you write full here again? This type of errors are everywhere in the manuscript, e.g in reporting ANC, ante natal care, antenatal care. Please revise for consistency. 

Results section of the abstract is too broad-please keep important findings only. 

I would recommend the author to revise this study abstract. 

Introduction 

This section needs substantial work. Information presented here is enough to write a good intro. However, what is lacking is cohesion. For instance, lines 70-71 and 72-75 are totally different concepts, though they were written together without proper link. Information presented in lines 85 and 86 look these factors are only the factors of exclusive breastfeeding, though this is not right. Similar mistakes are common everywhere. 

Methodsline 114, you analysed EDHS data, you did not conduct this study. 

Pease justification of selecting confounders. It seems you included all confounders without following any guidelines. I am also concerned about using the child comorbidity as confounders in this study. 

Results 

Please revise ways of presenting your results in the text.

Discussion 

This secretion is very poor and authors failed to offer effective policy. I would revise carefully to properly reflect this study's findings.

6. PLOS authors have the option to publish the peer review history of their article (what does this mean?). If published, this will include your full peer review and any attached files.

Reviewer #1: **Yes: **Aminur Rahman

Reviewer #2: No

Reviewer #3: No

---

## [Author Response · Author response to Decision Letter 0]

28 Oct 2020

Rebuttal letter Date: October/20/2020

PONE-D-20-23410

Exploring determinants of exclusive breast feeding among infants under- six months in Ethiopia using multilevel analysis

Erkihun Tadesse

PLOS ONE

Dear all,

We would like to thanks for these constructive, building and improvable comments on this manuscript that would improve substance and content of the manuscript. We considered each comments and clarification questions of editors and reviewers on the manuscript thoroughly. Our point-by-point responses for each comment and questions are described in detailed on the following pages. Further, the details of changes were shown by track changes in the supplementary document attached.

Editor questions/comments Response

Methodology

In the ethics statement in the Methods section and online submission information, please ensure that you have specified 

(1) whether consent was informed and 

(2) What type you obtained (for instance, written or verbal, and if verbal, how it was documented and witnessed). If the need for consent was waived by the ethics committee, please include this information.

-If you are reporting a retrospective study of medical records or archived samples, please ensure that you have discussed whether all data were fully anonymized before you accessed them and/or whether the IRB or ethics committee waived the requirement for informed consent. Thank you Editor for your constructive comment. We used the secondary data i.e. from 2016 Ethiopia Demographic health survey data for analysis. We amended the main manuscript file as well as in the online submission by adding: 

 All data used in this study were publicly available and aggregated secondary data with not having any personal identity. The data was fully available in the full DHS website (www.measuredhs.com).

Reviewer 1

1. Definition of EBF should be added except allow of oral re-hydration saline

 Thanks reviewer for your constructive comment. We have amended the definition of EBF in the main manuscript using the WHO recent definition; 

The World Health Organization (WHO) defines exclusive breastfeeding (EBF) as when 'an infant receives only breast milk, no other liquids or solids are given not even water, with the exception of oral rehydration solution, or drops/syrups of vitamins, minerals or medicines'

Line 165 the tense should not be future tense (will be) Thanks reviewer for your comments. We have replaced will be by was substituted by was. Since the aggregate value for all generated variables not normally distributed, it was categorized into groups based on the national median values and based on previous related studies.

3. Line 178 categorizing poverty is not enough; you need to mentioned how you have calculated the wealth index with what were the variables name Thank you for your comments. Poverty categorization is done according to wealth index. Wealth is the value of all natural, physical and financial assets owned by a household, reduced by its liabilities. Household wealth is a measure commonly used in food security assessments. It gives an idea of households’ ability to access food, the severity of food insecurity and gives information about the economic situation of the food insecure. It is used to differentiate between the poorer and the wealthier households in food security related indicators, such as food consumption, and thereby provides information on how to target the food insecure.

There are several ways in which wealth, economic status of households and living standards can be measured. Income, expenditure and consumption are three common measurements.

However, there are challenges in collecting and measuring income and expenditure accurately. An alternative is to use data on asset ownership and housing characteristics and combine this information into a proxy indicator such as the wealth index, which is created using principal component analysis (PCA). Asset ownership gives an indication of the longer-term economic status of a household and is less dependent on short-term economic changes compared with other wealth or poverty measures. 

The wealth index measures relative wealth and, unlike a poverty line, is not an absolute measure of poverty or wealth. When referring to the wealth of households based on the wealth index we can talk about poorer and wealthier households but we cannot conclude who is absolutely poor and wealthy. The wealth index quintiles divide the whole population into five equally large groups, based on their wealth rank. For example, in an area where only 10% of households fall below the poverty line, 40% of households will still fall into the two poorest quintiles and therefore be classified as the poorest.

The wealth index is commonly used in reports and analysis based on datasets from Demographic and Health Surveys (DHS) and is used to rank households into quintiles. The value of using the wealth index is especially recognized in contexts where reliable income and expenditure data is absent. The research questions related to the wealth index vary according to the different interests of the surveys. In DHS, the wealth index is chosen because of the major impact that wealth status has on household level health. It allows the investigator to identify the impact of wealth status on the research outcomes. DHS separates all interviewed households into five wealth quintiles to compare the influence of wealth on various population, health and nutrition indicators. The wealth index is presented in the DHS Final Reports and survey datasets as a background characteristic. 

Community level of poverty was generated by recoding wealth index categorized as: “<25% wealth index as high poverty”, 25%-50% wealth index as Moderate poverty level and “>50% wealth index as Low poverty level of the communities”. A woman is said to be; poorest, if she lives in the household with the first quintile (1st 20%) from the score of wealth index, poor; if she lives in the second quintile (2nd 20%), middle; if she lives in the household with the third quintile (3rd 20%), rich; if she lives in the household with the fourth quintile (4th 20%), richest; if she lives in the household with the fifth quintile (5th 20%) from the score of wealth index. 

4. Multi level logistic regression should be three outcome variables need to be mentioned in your write up?

 Thank you reviewer for the comment. In multinomial logistic regression there are three outcome variables but in multi-level logistic regression the outcome variable is binary (i.e. only two variables having dichotomous nature). Thus we considered multilevel logistic regression since the outcome is EBF having dichotomous nature (i.e. Yes if the infant is exclusively breast feed or not if the infant is not exclusively breast feed).

5. Birth weight: In survey measuring the birth weight is very uncommon. You should describe how you have collected birth weight. Thanks again for the comment. 

The Ethiopia demographic health survey data actually put birth weight of infants in grams, and it was categorized according to WHO criteria as low birth weight, normal and Macrosomia.

6. You should consider only 4 ANC that is the current indicator for global comparison

 Thanks reviewer for the comment. We have amended the document as: Receiving antenatal care at least four times increases the likelihood of receiving effective maternal health interventions during the antenatal period. This is one of the indicators in the Global Strategy for Women’s, Children’s and Adolescents’ Health (2016-2030) Monitoring Framework, and one of the tracer indicators of health services for the universal health coverage (SDG indicator 3.8.1) as well as an additional indicator in the Sustainable Development Goals framework. But using only the fourth ANC visit may not show either the women follow all 4 visits or not. 

7. Need to provide a multi-level logistic regression table otherwise description is not enough. Thanks again for the comment. The multi-level logistic regression table was presented in the main document at the end after references section.

Reviewer #2

In the method part, 

1.The authors should explain how they handle missing data. Thank you for the comment. To manage the missing data, complete case analysis (CCA) was used, only those cases with complete data on some set of variables.

Since the total missing number is very low i.e. tolerable (31(2.6%)) which is found from three factors/independent variables (child comorbidities=4, time of breast feeding initiation=24, ANC visit=3), the multilevel analysis by itself took the complete case analysis as default. And no missing in the outcome variable of interest.

2. Also they need to explain how they done wealth index or how the EDHS conducts wealth index. Thank you for your comments. 

Wealth is the value of all natural, physical and financial assets owned by a household, reduced by its liabilities. Household wealth is a measure commonly used in food security assessments. It gives an idea of households’ ability to access food, the severity of food insecurity and gives information about the economic situation of the food insecure. It is used to differentiate between the poorer and the wealthier households in food security related indicators, such as food consumption, and thereby provides information on how to target the food insecure.

There are several ways in which wealth, economic status of households and living standards can be measured. Income, expenditure and consumption are three common measurements.

However, there are challenges in collecting and measuring income and expenditure accurately. An alternative is to use data on asset ownership and housing characteristics and combine this information into a proxy indicator such as the wealth index, which is created using principal

component analysis (PCA). Asset ownership gives an indication of the longer-term economic status of a household and is less dependent on short-term economic changes compared with other wealth or poverty measures. 

The wealth index measures relative wealth and, unlike a poverty line, is not an absolute measure of poverty or wealth. When referring to the wealth of households based on the wealth index we can talk about poorer and wealthier households but we cannot conclude who is absolutely poor and wealthy. The wealth index quintiles divide the whole population into five equally large groups, based on their wealth rank. For example, in an area where only 10% of households fall below the poverty line, 40% of households will still fall into the two poorest quintiles and therefore be classified as the poorest.

The wealth index is commonly used in reports and analysis based on datasets from Demographic and Health Surveys (DHS) and is used to rank households into quintiles. The value of using the wealth index is especially recognized in contexts where reliable income and expenditure data is absent. The research questions related to the wealth index vary according to the different interests of the surveys. In DHS, the wealth index is chosen because of the major impact that wealth status has on household level health. It allows the investigator to identify the impact of wealth status on the research outcomes. 

DHS separates all interviewed households into five wealth quintiles to compare the influence of wealth on various population, health and nutrition indicators. The wealth index is presented in the DHS Final Reports and survey datasets as a background characteristic.

3-Besides, they need to explain community level factors in detail rather than simply describing in percentage.

 Thanks reviewer for the comment. 

Community level variables were generated by aggregating the individual characteristics in a cluster since EDHS did not collect data that can directly describe the characteristics of the clusters except place of residence. The aggregates were computed using the proportion of a given variables’ subcategory. Since the aggregate value for all generated variables not normally distributed, it will be categorized into groups based on the national median values and based on previous related studies. 

• Residence is the only community level variable neither computed nor relabeled. 

• Contextual region: Ethiopia is demarcated for administrative purpose into 11 regions, which are classified in to agrarian, pastoralist and city based according to the living status of the population. The regions Tigray, Amhara, Oromia, Southern Nations Nationalities and peoples (SNNP), Gambella and Benshangul Gumuz were categorized as agrarian. The Somali and Afar regions were grouped to form pastoralist region and the Harari region, Addis Ababa and Dire Dawa city administrations were grouped to form city based populations. This contextual region was created by recoding of the 11 codes/regions of the country in to 3 categories according to the predefined context as described above. 

• The other community level variables were created by aggregating the individual characteristics. Using cluster variable (v001) the variables to be generated for community level variables were kept, reordered and collapsed. Then the collapsed variables produce totals and percentages that were used to create their categories based on previously similar studies. Finally labeling and recoding accordingly were done to merge with the individual level variables for final regression analysis. 

Reviewer #3

Abstract 

- When I started reading this manuscript, I had a feeling that this study about individual and community level determinants of EBF. I am sure other readers will have the same feeling by seeing the study title. However, in actual practice, authors considered many determinants including individual level factors, household level factors, community level factors and child level factors. Thank you for your comments. Here we considered different level factors of EBF but all of them are bounded under individual level or community level for the sake of final analysis in two levels/clusters. The household level factors and child level factors were included under individual level factors of analysis. That is why we considered factors at individual level and community level.

-Line 32, you abbreviated exclusive breastfeeding before in the first line, why do you write full here again? Thanks for the comment. We described the word in short abbreviated form i.e. EBF and amended in the main manuscript.

Introduction 

 lines 70-71 and 72-75 are totally different concepts, though they were written together without proper link. 

- In lines 85 and 86 look these factors are only the factors of exclusive breastfeeding, though this is not right. Thanks again for the comment. We have amended by presenting them in separate paragraph in the manuscript file.

These are not factors affecting EBF but we mentioned here some of the significant factors from previous litreatures. And amended in the main manuscript. 

Even if the government of Ethiopia tried to promote and implement exclusive breast feeding practice for infants under six months of age but still there is poor improvement. This could be several contributing factors including Low level of educational status, maternal employment, operative delivery, late initiation of breast feeding, low antenatal and postnatal visits, increased production of formula foods and poor counseling of mother regarding EBF. Furthermore, Exclusive breast feeding was also influenced by community level factors including place of residence, contextual region, community level of maternal education, community level of ANC and PNC utilization, community level of employment, and community level of media exposure. 

Methods 

-line 114, you analyzed EDHS data; you did not conduct this study. Thank you for your comment. A study is conducted using primary or secondary data. In this research, we are using secondary data from the 2016 Ethiopia Demographic health survey data where data collected from the whole parts of the country i.e. nine regions and two city administrations. Thus, description of the study area where the data collected is one of the recommended protocols in research. That is why we described the study setting i.e. Ethiopia in the main manuscript.

-Pease justification of selecting confounders. It seems you included all confounders without following any guidelines. I am also concerned about using the child comorbidity as confounders in this study. Multivariable analysis is one of the methods of confounders controlling mechanism at analysis phase. At analysis phase, Bi-variable multilevel logistic analysis was computed and p-value up to 0.2 was selected to fit multi-variable multilevel logistic regression model. Those variables with p values less than or equal to 0.2 were taken for multivariable analysis to control confounding. Finally, in this study age of the infants, sex of the infants, infant comorbidities, wealth index, antenatal care visit, contextual region, community postnatal care visit and community level of employment status were statistically significant variables with exclusive breastfeeding in the final model using multivariable analysis by controlling confounding. 

 Child comorbidity is not a confounder rather it is truly associated with the outcome variable when the multivariable regression was performed. 

 Results 

Please revise ways of presenting your results in the tex. Thanks for the comment. According to your query We made an exhaustive revision in presenting the result section. 

Discussion 

-This section is very poor and authors failed to offer effective policy. Thanks for the comment. According to your query we discussed our finding in detail. We also provided the strength and limitation of this study in detail. Furthermore the implication of the study was described clearly in detail at the end.

Implications of the study

Exclusive breastfeeding is one of the core indicators of infant and young child feeding, among strategies of reducing infant mortality and morbidity. It determines future growths and developments of the infants both in physical and mental. The study demonstrates that addressing individual and community level factors associated with exclusive breast feeding practice through policies and programs was essential to improve exclusive breastfeeding practice. Therefore, emphasis should be given on encouraging women to have ANC and PNC follow ups, where they may get information and education that will improve exclusive breastfeeding. The infants with comorbid conditions need attention since they have low exclusive breastfeeding tendency. Since maternal employment and wealth status have higher likelihood of exclusive breastfeeding, empowering women both economically and in their employment status is implicated. Therefore, interventions on individuals and community levels were demanded for saving lives of the infants and reduction of economic losses of a country.

---

## [Decision Letter · Decision Letter 1]

22 Dec 2020

Exploring the Determinants of Exclusive Breastfeeding among Infants under-six Months in Ethiopia Using Multilevel Analysis

PONE-D-20-23410R1

Dear Dr. Tadesse,

We’re pleased to inform you that your manuscript has been judged scientifically suitable for publication and will be formally accepted for publication once it meets all outstanding technical requirements.

Kind regards,

Mohammad Rifat Haider, MBBS, MHE, MPS, PhD

Academic Editor

PLOS ONE

Additional Editor Comments (optional):

Reviewers' comments:

Reviewer's Responses to Questions

**Comments to the Author**

1. If the authors have adequately addressed your comments raised in a previous round of review and you feel that this manuscript is now acceptable for publication, you may indicate that here to bypass the “Comments to the Author” section, enter your conflict of interest statement in the “Confidential to Editor” section, and submit your "Accept" recommendation.

Reviewer #1: (No Response)

Reviewer #2: All comments have been addressed

2. Is the manuscript technically sound, and do the data support the conclusions?

Reviewer #1: Yes

Reviewer #2: Yes

3. Has the statistical analysis been performed appropriately and rigorously? 

Reviewer #1: Yes

Reviewer #2: Yes

4. Have the authors made all data underlying the findings in their manuscript fully available?

Reviewer #1: Yes

Reviewer #2: Yes

5. Is the manuscript presented in an intelligible fashion and written in standard English?

Reviewer #1: Yes

Reviewer #2: Yes

6. Review Comments to the Author

Reviewer #1: I feel that the authors have written a good manuscript and that the paper is acceptable for publication.

Reviewer #2: I would like to thank the authors for considering all my comments and shaping the manuscript as per comment given. I believe that a manuscript is now in a position for publication.

7. PLOS authors have the option to publish the peer review history of their article (what does this mean?). If published, this will include your full peer review and any attached files.

Reviewer #1: **Yes: **Aminur Rahman

Reviewer #2: **Yes: **Negeso Gebeyehu Gejo

---

## [Editor Report · Acceptance letter]

29 Dec 2020

PONE-D-20-23410R1 

Exploring the Determinants of Exclusive Breastfeeding among Infants under-six Months in Ethiopia Using Multilevel Analysis 

Dear Dr. Tadesse Amsalu:

I'm pleased to inform you that your manuscript has been deemed suitable for publication in PLOS ONE. Congratulations! Your manuscript is now with our production department. 

Kind regards, 

on behalf of

Dr. Mohammad Rifat Haider 

Academic Editor

PLOS ONE